# The human health burden of non-typhoidal *Salmonella enterica* and *Vibrio parahaemolyticus* foodborne gastroenteritis in Shanghai, east China

Yan Chen[1]*, Hong Liu[2], Min Chen[2], He-Yang Sun[1], Yong-Ning Wu[1]

**1** NHC Key Laboratory of Food Safety Risk Assessment, Chinese Academy of Medical Science Research Unit, China National Center for Food Safety Risk Assessment, Beijing, People's Republic of China,
**2** Shanghai Municipal Center for Disease Control and Prevention, Shanghai, People's Republic of China

☯ These authors contributed equally to this work.
* chenyan@cfsa.net.cn

**Data Availability Statement:** All relevant data are within the manuscript and its Supporting information files.

## Abstract

Information on the burden of disease due to foodborne pathogens in China is quite limited. To understand the incidence of foodborne gastroenteritis due to non-typhoidal *Salmonella enterica* and *Vibrio parahaemolyticus*, population survey and sentinel hospital surveillance were conducted during July 2010 to June 2011 in Shanghai, east China, and a model for calculating disease burden was established. The multiplier for gastroenteritis caused by these pathogens was estimated at 59 [95% confidence interval (CI) 30–102]. Annual incidence per 100,000 population in Shanghai was estimated as 48 (95% CI 24–83) and 183 (95% CI 93–317) cases for foodborne non-typhoidal salmonellosis and *V. parahaemolyticus* gastroenteritis, respectively, illustrating that bacterial gastroenteritis due to these two pathogens poses a substantial health burden. There is a significant difference between our simulated incidence and the data actually reported for foodborne diseases, indicating significant underreporting and underdiagnosis of non-typhoidal *S. enterica* and *V. parahaemolyticus* gastroenteritis in the surveillance area. The present research demonstrates basic situation of the health burden caused by major foodborne pathogens in the surveillance area. Enhanced laboratory-based sentinel hospital surveillance is one of the effective ways to monitor food safety in east China.

## Introduction

Foodborne disease is an important public health issue worldwide, and it is also China's number one food safety issue, causing a huge burden of disease and major economic losses [1–7]. Effective control of foodborne diseases needs to be based on information from foodborne disease burden assessments, but the true burden of foodborne diseases in China is not cleared yet. Chinese foodborne disease passive surveillance systems mainly collect information on outbreaks and rarely collect information on sporadic diseases [8]. According to the Management

**Funding:** This project was supported by grant from the National Natural Science Foundation of China (No. 81673175).

**Competing interests:** The authors have declared that no competing interests exist.

of Food Poisoning Incidents released in 1999: "Units where food poisoning or suspected food poisoning incidents occurred and those that receive food poisoning or suspected food poisoning patients for treatment should report the relevant information to the local health administrative department in a timely manner [9]." Since only a small fraction of foodborne diseases occur relative to identified outbreaks, reported foodborne disease through the passive surveillance system is often insufficient [10]. To overcome the inherent defects of passive surveillance, the United States [2, 3, 11], the United Kingdom [4, 12], the Netherlands [6], Australia [5], Canada [13], Jordan [14], and Japan [15], have conducted research to better estimate the burden of foodborne disease and foodborne pathogens more accurately, however, only one study was conducted in south China [16], and such studies were lacking in east China.

Estimating the burden of disease for specific foodborne pathogens is necessary for precisely estimating foodborne disease burden and monitoring the impact of management measures to prevent and control foodborne disease. Assessing the burden of foodborne disease is complicated because it is difficult to determine whether the disease is absolutely associated with food. Acute gastrointestinal illness (AGI) is a common form of foodborne disease. Research ascertaining the burden of AGI and the proportion of foodborne pathogens among patients provides important basic data for estimating foodborne disease burden [17]. Studies to estimate community pathogen-specific incidence can be of two designs: (1) community-based AGI cohort study [6, 18], and (2) estimate the monitoring data multiplier by using additional data [14–16, 19, 20].

For specific and effective control measures, it is necessary to determine foodborne disease severity. From 2010 to 2011, the Shanghai Municipal Center for Disease Control and Prevention launched a pilot project for active surveillance of foodborne diseases. Non-typhoidal *Salmonella enterica* and *Vibrio parahaemolyticus* are the most common bacteria that cause foodborne disease outbreaks in China [8]. Using data from cross-sectional population survey on AGI, along with sentinel hospital surveillance on specific foodborne pathogens, the health burden of foodborne gastroenteritis in Shanghai of east China caused by these two pathogens was estimated.

## Materials and methods

### Population survey

To estimate the incidence of AGI in the general population, a twelve-month, face-to-face survey from July 2010 through June 2011 of randomly selected respondents were conducted in Shanghai of east China [7]. The sentinel sites were: Luwan District (population 248,779) and Qinpu District (population 1,081,022). Population in the sentinel sites represents about 5.8% of the total Shanghai permanent resident population (23,019,196) in 2010. Each sentinel site was divided into several blocks according to population proportioned distribution. A total of 251 blocks were generated from the two sentinel sites. Households were randomly selected from each block and the number of households surveyed was proportional to the population size. Within each household, the individual who was next to celebrate his/her birthday was chosen to participate in the survey. Written and informed consent was received from all respondents and parents or guardians of the minors prior to the interview. Proxy respondents were applied for people aged less than 12 and aged between 12–18 in which it was up to the parents or guardians to decide.

AGI was defined as diarrhea of three and above loose stools during any 24-h period or significant vomiting with at least one other symptom, such as fever, abdominal cramps or pain, but excluding those persons who reported their symptoms of diarrhea or vomiting to be due to non-infectious causes such as Crohn's disease, irritable bowel syndrome, colitis,

diverticulitis of large intestine, pregnancy, excess alcohol, chemotherapy/radiotherapy, drugs, or food allergy. The proportion of AGI cases seeking medical care, and the proportion of cases submitting a stool specimen for testing was also determined. Through weighting age, gender, and residence, differences in estimates for survey and target populations were adjusted using the 2010 Shanghai census data [21]. Questionnaire data were entered and analyzed using Epi-Data version 3.1 (EpiData Association, Odense M, Denmark) and SPSS version 16.0 (SPSS Inc., Chicago, IL, USA).

## Sentinel hospital surveillance

During the same period of the population survey, hospital surveillance was carried out at eight sentinel hospitals, among which two were secondary hospitals and six primary hospitals. There was a total of two tertiary hospital, four secondary hospitals and 14 primary hospitals in the surveillance area during the period of 2010 and 2011. A primary hospital was defined as a community hospital that provided primary health services; a secondary hospital was defined as a local hospital that provided comprehensive health services; and a tertiary hospital was defined as a regional hospital that provided comprehensive and specialized health services [22]. Rectal swabs or stool specimens from patients with diarrhea were collected from the sentinel hospitals, and all specimens were tested for non-typhoidal *S. enterica* and *V. parahaemolyticus* by the laboratories of the local Center for Disease Control and Prevention. We assumed that the stool samples sent to the hospital laboratories were representative of the community samples.

For *S. enterica* detection, specimens were enriched in selenite broth, followed by surface plating (or plating) on Bismuth sulfite agar and xylose-lysine-desoxycholate (XLD) agar (or Hektoen enteric agar, CHROMagar *Salmonella* agar) [23]. With reference to *V. parahaemolyticus* detection, specimens were enriched in alkaline peptone water, followed by surface plating (or streaking) on thiosulfate citrate bile salts sucrose agar or CHROMagar *Vibrio* agar [24]. The plates were incubated at 37°C for 18–24 hours. We defined a case of diarrhea as a person with three and above loose stools during any 24-h period.

## Burden of disease calculation

We multiplied the sentinel hospital surveillance area population, AGI incidence per person-year, the proportion of cases seeking medical care (the inverse of Multiplier One), and the proportion of cases submitting a stool specimen for testing among those seeking medical care (the inverse of Multiplier Two) to estimate the number of stool specimens submitted in the surveillance area. Then, the number of submitted stool specimens was divided by the number of stool specimens tested to estimate the proportion of laboratory pathogen testing. Multiplier Three is the inverse of the proportion of laboratory pathogen testing.

According to proficiency testing program in Guangdong province, the non-typhoidal *S. enterica* isolation sensitivity rate of the laboratories was 87.5% [16]. Therefore, it was assumed that the laboratory testing sensitivity in Shanghai was 87.5%, ranging from 85% to 100%. Multiplier Four is the inverse of the proportion of laboratory identifying pathogen.

The above mentioned multipliers were multiplied to estimate the multiplier for these surveillance artifacts (Multiplier Total, $M_T$). Fig 1 shows the pyramid of the burden of pathogen-specific gastroenteritis. To account for uncertainty, multipliers were modeled using the Pert distribution (Table 1) [25]. The estimation model was performed using @RISK (version 7.6, Palisade, Newfield, N.Y.), with 25,000 iterations per estimation. The number of positive specimens tested was multiplied with $M_T$ to estimate the number of positive samples in the

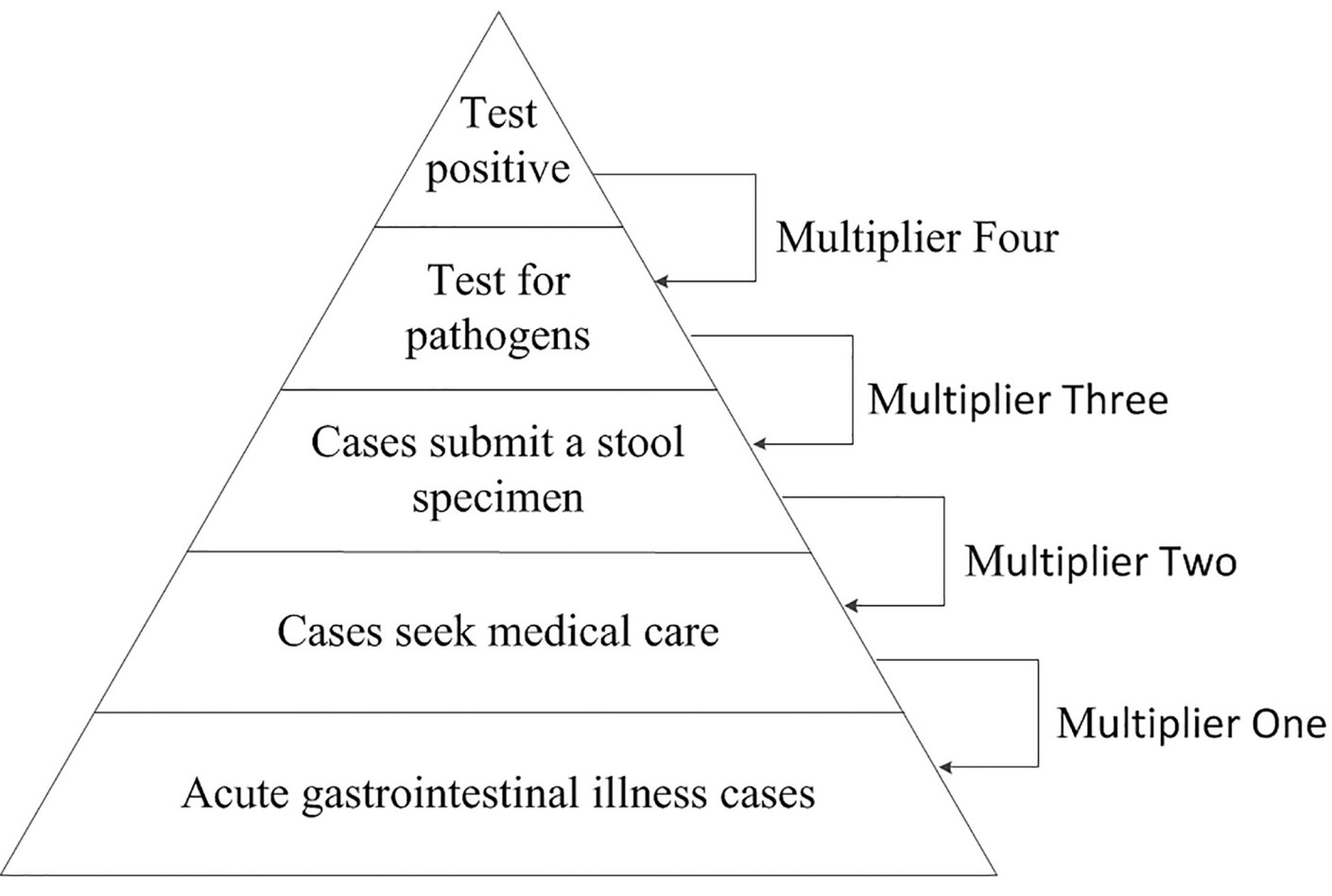

**Fig 1. Pyramid of the burden of pathogen-specific gastroenteritis.**

surveillance area. By using the population of the surveillance area reported in the 2010 census, the estimated annual disease per 100,000 population in the surveillance area were assessed.

Since there are no data on the proportion of foodborne pathogen-specific transmission in China, the proportions were based on data published by Hald et al. for WPR B region (including China) and were multiplied by the number of diseases to obtain foodborne non-typhoidal salmonellosis [26], while the proportion based on US research was used to obtain foodborne *V. parahaemolyticus* gastroenteritis [2]. The estimated pathogen-specific numbers

**Table 1. Multipliers used to determine health burden of non-typhoidal salmonellosis and *Vibrio parahaemolyticus* gastroenteritis in Shanghai, east China, 2010–2011.**

| Multiplier[a] | Surveillance step | Distribution |
|---|---|---|
| Multiplier One ($M_1$) | Cases seeking medical care | Pert (2.1, 2.7, 3.8) |
| Multiplier Two ($M_2$) | Cases submitting a stool specimen for testing | Pert (1.9, 2.9, 5.9) |
| Multiplier Three ($M_3$) | Laboratory performing test for pathogens | Pert (2.9, 6.0, 8.9) |
| Multiplier Four ($M_4$) | Laboratory identifying pathogens | Pert (1.0, 1.1, 2.0) |
| Multiplier Total ($M_T$) | The multiplier for the total surveillance artifacts | $M_1{}^*M_2{}^*M_3{}^*M_4$ |

[a] The multiplier for each surveillance step is the inverse of the proportion responding positively.

of foodborne disease cases were compared with the routinely reported numbers of foodborne outbreak cases so as to calibrate the surveillance data [8].

### Scientific ethics

The study protocol on population survey and sentinel hospital surveillance was approved in 2010 by the Committee on Human Experimentation of the National Institute for Nutrition and Food Safety, Chinese Center for Disease Control and Prevention. The ethics committee waived the requirement for informed consent from patients with diarrhea.

## Results

### Population survey

Between July 2010 and June 2011, 7,176 persons were interviewed (response rate = 99.4%). Distribution of demographic characteristics of residents and survey respondents are shown in Table 2. In general, survey respondents were more likely to be female, older, less educated, larger household size and more likely to live in the rural area than residents. Of the 7,176 persons included in the survey, 108 (1.5%) reported having experienced symptoms of gastro-enteritis in the four weeks prior to interview. Of these respondents, eight were declared due to non-infectious causes and included in the non-case group, leaving 100 respondents to be

**Table 2. Distribution of demographic characteristics of residents and survey respondents.**

| Variable | Proportion of Residents (%) | Proportion of Survey Respondents (%) |
|---|---|---|
| **Gender** | | |
| Male | 51.5 | 48.2 |
| Female | 48.5 | 51.8 |
| **Age (years)** | | |
| 0–4 | 3.4 | 0.5 |
| 5–14 | 5.2 | 1.6 |
| 15–24 | 16.3 | 4.3 |
| 25–44 | 36.9 | 16.6 |
| 45–64 | 28.1 | 44.3 |
| $\geq$ 65 | 10.1 | 32.7 |
| **Education** | | |
| Preschool children | 4.1 | 0.8 |
| Illiterate | 3.0 | 11.4 |
| Primary | 13.6 | 23.4 |
| Secondary | 36.5 | 48.1 |
| Tertiary | 21.0 | 10.6 |
| University | 21.9 | 5.7 |
| **Household size (number of person)** | | |
| 1–2 | 51.5 | 41.5 |
| $\geq$ 3 | 48.5 | 58.5 |
| **Household type** | | |
| No residents < 18 years | n.a. | 77.7 |
| At least one resident < 18 years | n.a. | 22.3 |
| **Residence** | | |
| Urban | 89.3 | 54.3 |
| Rural | 10.7 | 45.7 |

maintained in the case group. The prevalence of AGI in the four weeks prior to interview, adjusted for age, gender, and residence, was 1.2% [95% confidence interval (CI) 1.0–1.5]. This represents an average of 0.16 (95% CI 0.15–0.17) occurrences of AGI per person-year in Shanghai. Among AGI cases, 36.9% (95% CI 26.5–47.2) visited a doctor, and among them 34.4% (95% CI 17.0–51.9) submitted a stool specimen.

### Sentinel hospital surveillance

Between July 2010 and June 2011, a total of 4,568 patients with diarrhea presented to a hospital participating in surveillance, among which, 4,548 swab/stool specimen were collected and tested. Non-typhoidal *S. enterica* and *V. parahaemolyticus* were isolated from 19 (0.4%) and 48 (1.1%), respectively. The month-wise isolation of these pathogens varied between 0–1.7% and 0–3.2%, respectively (Fig 2), while more information regarding the two studied bacterial species was not collected. No detailed personal information was collected on patients with diarrhea.

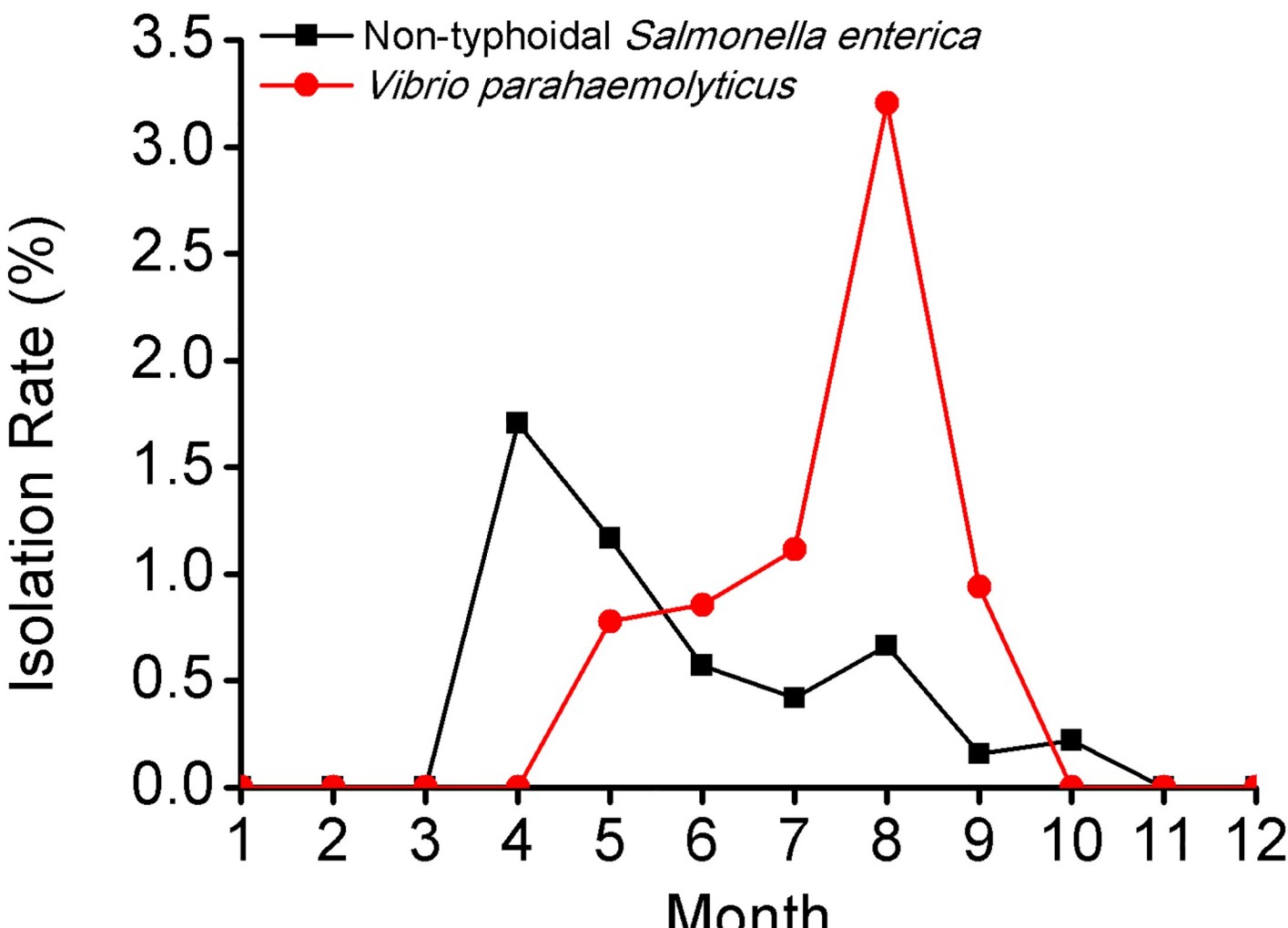

**Fig 2. Month-wise isolation rate for non-typhoidal *Salmonella enterica* and *Vibrio parahaemolyticus* in Shanghai, east China, 2010–2011.**

**Table 3. Steps for the calculation of the frequency of laboratory performing test for pathogens in the surveillance area in Shanghai, east China, 2010–2011.**

|  | Mean (95% CI) |
|---|---|
| AGI[a] incidence per person-year | 0.16 (0.15–0.17) |
| AGI episodes per year (*n*) | 212,768 (199,470–226,066) |
| Proportion of cases seeking medical care (%) | 36.9 (26.5–47.2) |
| Medical consultations for AGI per year (*n*) | 78,511 (56,384–100,427) |
| Proportion of cases submitting a stool specimen (%) | 34.4 (17.0–51.9) |
| No. stool specimens submitted | 27,008 (13,347–40,747) |
| No. stool specimens tested | 4,548 |
| Proportion of laboratory pathogen testing (%) | 16.8 (11.2–34.1) |

[a] AGI—acute gastrointestinal illness.

## Burden of disease calculation

According to data from the National Bureau of Statistics, the population in the surveillance area in 2010 was 1,329,801. By multiplying the incidence of AGI 0.16 per person-year, the AGI episodes per year in the surveillance area was estimated to be 212,768. Among persons with AGI, 36.9% sought medical care. Extrapolating this figure to the population of surveillance area indicates that 78,511 persons with AGI sought medical care. Among people seeking medical care, 34.4% submitted a stool specimen for testing. Extrapolating this figure to the population of surveillance area indicates that 27,008 individuals with AGI submitted stool specimens (2,031 specimens/100,000 population). Considering that the number of stool specimens tested in the surveillance area was 4,548, the frequency of laboratory performing test for pathogens was estimated to be 16.8% (Table 3). For each person with laboratory confirmed salmonellosis or *V. parahaemolyticus* gastroenteritis there were 59 (95% CI 30–102) infected persons in the community ($M_T$).

The estimated cases in the surveillance area during the 12-month study were 1,121 cases (95% CI 570–1,938) of salmonellosis and 2,832 cases (95% CI 1,440–4,896) of *V. parahaemolyticus* gastroenteritis (Table 4). Considering the population of the surveillance area, the annual incidence estimated by the model developed herein was 84 (95% CI 43–146) cases for salmonellosis and 213 (95% CI 108–368) cases for *V. parahaemolyticus* gastroenteritis per 100,000 population. Annual incidence per 100,000 population in Shanghai was estimated as 48 (95% CI 24–83) and 183 (95% CI 93–317) cases for foodborne non-typhoidal salmonellosis and *V. parahaemolyticus* gastroenteritis, respectively.

## Discussion

This is the first disease burden report for foodborne pathogens in east China. Using data from a population survey and sentinel hospital surveillance, the burden of foodborne gastroenteritis

**Table 4. Estimated health burden of non-typhoidal salmonellosis and *Vibrio parahaemolyticus* gastroenteritis in Shanghai, east China, 2010–2011.**

| Pathogen | No. positive specimens | Estimated positive specimens in the surveillance area (95% CI) | Estimated disease cases per 100,000 population (95% CI) | Estimated percentage of foodborne transmission | Estimated foodborne disease cases per 100,000 population (95% CI) | Reported foodborne disease outbreak cases per 100,000 population |
|---|---|---|---|---|---|---|
| Non-typhoidal *Salmonella enterica* | 19 | 1,121 (570–1,938) | 84 (43–146) | 57% | 48 (24–83) | 0.08 |
| *Vibrio parahaemolyticus* | 48 | 2,832 (1,440–4,896) | 213 (108–368) | 86% | 183 (93–317) | 1.71 |

in Shanghai was estimated to be 11,061 (95% CI 5,624–19,122) cases of salmonellosis and 42,159 (95% CI 21,437–72,886) cases of *V. parahaemolyticus* gastroenteritis. This indicates that AGI caused by these two pathogens poses a substantial burden in the Shanghai population. While knowing the absolute number of community cases may not be necessary to identify an outbreak, this information is very important for decision makers because it is useful both for formulating public health policies and for estimating the cost of disease.

Our estimate of foodborne salmonellosis (48 cases per 100,000 population) was lower than those estimated in Japan (199 per 100,000) [15], the United Kingdom (220 per 100,000) [27], the United States (340 per 100,000) [2], and Australia (422 per 100,000) [5]. Compared with other data related to China, the foodborne salmonellosis incidence estimated in the present study was lower than that reported for Guangdong province (392 per 100,000) [16], as determined from a literature review (627 per 100,000) [28], and that reported in a global burden study (3,600 per 100,000) [29].

*V. parahaemolyticus* has been the leading cause of foodborne disease outbreaks in China [8]. Our estimate of foodborne *V. parahaemolyticus* gastroenteritis (183 per 100,000) was much higher than those estimated in the United Kingdom (<1 per 100,000) [27], Australia (4 per 100,000) [5], the United States (12 per 100,000) [2], and Japan (65 per 100,000) [15]. Shanghai is a coastal province, and previous data reported that the number of foodborne outbreaks of *V. parahaemolyticus* in this province ranks first in China [30]. Shanghai's health administration needs to take effective measures to strengthen the prevention and control of foodborne gastroenteritis caused by *V. parahaemolyticus*.

Diseases are divided into outbreak and sporadic forms. When designing and implementing pathogen-specific foodborne disease burden studies, it is necessary to consider the distinction between outbreak and sporadic cases. As shown by data from Japan and the United States, *S. enterica* and *V. parahaemolyticus* infections are mainly sporadic, and outbreaks are less common [2, 15]. Because outbreaks and sporadic diseases have similar case characteristics [31], the analysis of the estimated disease burden based on laboratory-confirmed sporadic cases adopted in this study is feasible. In the present study, the estimates of *S. enterica* and *V. parahaemolyticus* were remarkably higher than the numbers reported to passive surveillance as foodborne disease outbreak cases, findings similar to those reported in Japan and the United States, highlighting the fact that many foodborne diseases are not captured in the present passive outbreak surveillance system [2, 15].

It should be recognized that the uncertainty of this study falls within the range of other countries' estimates [2, 12, 14–16]. For each culture-confirmed case, we estimated that there were 59 cases of salmonellosis or *V. parahaemolyticus* gastroenteritis in the community. These differences in uncertainty of the study may be mainly due to methodological differences, such as laboratory practices and surveillance systems. In 2011, the China National Center for Food Safety Risk Assessment officially initiated the National Laboratory-based Foodborne Disease Surveillance Network, which includes population surveys and active sentinel surveillance, aiming at estimating the burden of foodborne diseases [32]. However, due to the large number of AGI cases in China, the limited number of existing laboratory monitoring specimens will inevitably cause high uncertainty, resulting in inaccurate estimates of foodborne disease burden. Although cohort studies are more complex and costly, they can provide much more accurate community-specific rates of pathogen-specific infections when combined with laboratory tests of case specimens. In order to more accurately determine and prioritize food safety issues in China, and to evaluate and quantify the burden of foodborne diseases outbreaks, it is necessary to establish a cohort study of acute gastroenteritis including case-test specimens in order to obtain a more accurate incidence of major foodborne pathogen infections, so as to evaluate the burden of foodborne disease.

Other sources of error in the present study may also have affected the estimates. First, we use the proportion of cases submitting a stool specimen from population surveys. In China, people submit stool specimens more for leukocyte testing than for specific pathogens. Thus, we may have overestimated the proportion of persons infected with pathogens who provided stool specimens for culture and, therefore, we may have underestimated the multiplier for this step. Second, we also estimated the rate of laboratory testing sensitivity. Further studies are needed to confirm this estimate, if the actual sensitivity is higher, then our calculations overestimate the burden of disease. Finally, the estimation of community prevalence of infection with individual pathogens from the testing of a relative lower proportion of stool samples from cases is likely to be subject to bias and considerable uncertainty. In the present study, the stool samples were obtained from patients who sought medical care for AGI. AGI visits may be the result of a number of factors, such as urban/rural differences, which could bias whether these samples are representative of the community samples.

When a person is exposed to a foodborne pathogen, their peak response falls on a continuum from no infection to infection (asymptomatic) to disease (symptomatic) to severe disease (hospital) to death. Where the peak response falls on this continuum depends on the outcome of the interaction between the pathogen, host, and food (disease triangle). Consequently, if two communities had the same disease rate but one community had more high-risk individuals, the disease rate would be a poor indicator of health burden because the severity of disease would higher in the community with more high-risk individuals. Therefore, it is necessary to collect data on severity of disease in future researches, and the current assessment could be improved by estimating the health burden that takes severity of disease into account.

## Conclusions

In conclusion, the estimated large number of salmonellosis and *V. parahaemolyticus* gastroenteritis cases occurring every year in the surveillance area indicates that these two pathogens pose a substantial health burden in Shanghai, east China. After considering the differences among distinct pathogens, these methods can also be applied in a similar manner to assess the burden of additional foodborne pathogens. Ongoing studies on the economic burden of major foodborne pathogen infections are necessary to more accurately assess the burden of foodborne disease in China. China should further strengthen global cooperation in the field of foodborne disease burden assessment in order to provide improved data support for global burden estimation of foodborne diseases.

## Supporting information

**S1 Questionnaire.**
(DOC)

## Acknowledgments

The authors acknowledge all of the collaborators involved in the population survey and sentinel hospital surveillance for providing the data essential to this analysis.

## Author Contributions

**Formal analysis:** Yan Chen.

**Funding acquisition:** Yan Chen.

**Investigation:** Hong Liu, Min Chen.

**Methodology:** Yan Chen.

**Project administration:** Yan Chen.

**Supervision:** Yong-Ning Wu.

**Writing – original draft:** Yan Chen, He-Yang Sun.

**Writing – review & editing:** Yan Chen.

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
