## [Decision Letter · Decision Letter 0]

24 Aug 2020

PONE-D-20-22153

The Human Health Burden of nontyphoidal Salmonella and Vibrio parahaemolyticus Foodborne Gastroenteritis in Shanghai, East China

PLOS ONE

Dear Dr. Chen,

Thank you for submitting your manuscript to PLOS ONE. After careful consideration, we feel that it has merit but does not fully meet PLOS ONE’s publication criteria as it currently stands. Therefore, we invite you to submit a revised version of the manuscript that addresses the points raised during the review process.

The Academic Editor would like to apologize to the authors for the long duration of the review process; it was due to the unavailability of multiple reviewers with expertise in the field of surveillance and epidemiology who were invited by the editor to handle the manuscript but could not accept the invitation. Review comments have been timely provided by only one invited reviewer and are attached in the present letter. In order to provide the authors with a timely decision, this manuscript proceeded to the next stage of the editorial process, based on the evaluation provided by one external reviewer and the academic editor herself.

Editor’s comments

Although the Academic Editor is not an epidemiologist, raised comments regarding the overall approach of the presented study from a food microbiology perspective, its scientific soundness as well as the quality of the manuscript are summarized below. According to the editor’s opinion, the manuscript should be subject to minor revisions before its publication in PLOS ONE is considered. Beyond the more specific comments that are following, a general a comment is an overall cross-check with regard to English grammar/syntax (maybe a review by an English expert/native speaker should be considered).  

- Title: both in the manuscript’s title and throughout the manuscript, please use the species name “*Salmonella enterica*” (or S. enterica after first mentioning) instead of the genus name (i.e. *Salmonella*). The genus Salmonella includes two species (i.e. *Salmonella enterica *and *Salmonella bongori*) and only *S. enterica* is regarded as a foodborne pathogen for humans.

- L24-25: please revise to “…cases for non-typhoidal salmonellosis and *V. parahaemolyticus* infection, respectively, illustrating that bacterial gastroenteritis due to these two pathogens poses a substantial health burden”

- L27: revise to “…and the data actually reported for foodborne diseases…”

- L45: I would suggest using “relative” instead of “proportionate”

- L48: please correct to “the Netherlands”

- L53: please revise to “…for precisely estimating foodborne disease burden and monitoring the impact…”

- L58-59: please revise to “…among patients provides important basic data for estimating foodborne disease burden”

- L64: the phrase “foodborne pathogenic infections” is rather specific for such a general statement; I would suggest using “foodborne illness severity” instead.

- L92: correct to “eight sentinel hospitals”

- L94-95: please correct to “Rectal swabs or stool specimens from patients with diarrhea were collected from the sentinel hospitals…”

- L96: revise to “non-typhoidal S. enterica and V. parahaemolyticus by the laboratories of the Chinese Center for Disease Control and Prevention”; in general, be consistent using the same single (and not multiple) terms throughout the manuscript, e.g. “non-typhoidal” (instead of “non-typhoid”  or “nontyphoidal”), “foodborne” (instead of “food-borne”), etc.

- L98: please correct to “…were representative of the community samples”

- L99-103: revise to “For *S. enterica* detection, specimens were enriched in selenite broth, followed by surface plating (or plating) on Bismuth sulfite agar and xylose-lysine-desoxycholate (XLD) agar. With reference to *V. parahaemolyticus* detection, specimens were enriched in alkaline peptone water, followed by surface plating (or streaking) on thiosulfate citrate bile salts sucrose agar or CHROMagar *Vibrio* agar”. Moreover, were specific ISO protocols applied for these selective enrichment procedures? If so, please refer to the specific protocols (as in-text citations with the corresponding additions in the references’ list).     

- L107-113: it would be better for the reader (particularly a non-expert in epidemiology) if definitions of M1, M2 and M3 are provided in brief

- L111: please change to “Then, the number of submitted stool specimens was divided by…”

- L112: by “laboratory performing test for pathogens” you refer to “laboratory pathogen testing”? If so, consider using the second (simpler) term both here and wherever else applicable in the manuscript’s text.

- L114: change to “It was assumed that the laboratory testing sensitivity was 87.5%...”; in general, use third person (and not “we”) when providing descriptions pertinent to the Materials and Methods and/or Results section of the manuscript.

- L117: please revise to “Figure 1 shows the pyramid of the burden of pathogen-specific foodborne gastroenteritis”

- L118-119: please revise to “…were modeled using the Pert distribution. The estimation model was developed using @RISK…”

- L120: please revise to “The number of positive specimens tested was multiplied with M to estimate the number of sportive samples in the surveillance area”

- L123: please correct to “…were assessed”

- L124: revise to “Pyramid of the burden of pathogen-specific gastroenteritis”. Also, is this the appropriate place in the manuscript’s text for a figure caption (and the same applies for Fig. 2 caption in L157-158)?

- L125-126: please revise to “…non-typhoidal salmonellosis and Vibrio parahaemolyticus infection…”; please use the same phrase when referring to the infection (gastroenteritis) caused by these two pathogens throughout the manuscript for simplification and consistency reasons.

- Table 1: in the last line (referring to M4) and the last column (referring to the distribution) please use a decimal digit for all three numbers (namely, 1.0, 1.1 and 2.0).

- L129-135: please revise as following recommended: “Since there are no data on the proportion…on data published by Hald et al. [23]…and were multiplied by the number of illnesses to obtain foodborne non-typhoidal salmonellosis [23], while the proportion based on US research was used to obtain…The estimated pathogen-specific numbers of foodborne illness cases were compared with the routinely reported numbers of foodborne outbreak cases so as to calibrate the surveillance data”. Please keep in mind that “data” is plural (singular form the Latin word “datum”), and proceed with all required grammar corrections when using this term throughout the manuscript.

- L138: change to “…the National Institute for Nutrition and Food Safety…”

- L149-150: please revise to “…36.9% (95% CI 26.5-47.2) visited a doctor, and among them 34.4% (95% CI 17.0-51.9) submitted a stool specimen”

- L155-156: please correct to “…(Fig. 2), while more information regarding the two studied bacterial species was not collected”

- L157-158: please revise the Figure 2 caption to “Month-wise isolation rate for non-typhoidal Salmonella enterica and Vibrio parahaemolyticus in Shanghai, east China, 2010-2011”

- L165: please correct to “Among people seeking medical care, 34.4%...”

- L168: please correct to “Considering that the number of stool specimens tested in the surveillance area was 4,548…”

- L170: change to “…laboratory confirmed salmonellosis or V. parahaemolyticus infection…”; also to what does “respectively” refer to? (a single percentage, namely 71%, is mentioned herein).

- Table 2: I would suggest moving the percentage symbol (%) in the first (from the second) column of the table in parentheses (wherever applicable), e.g., “AGI incidence per person-year (%)”. Moreover, spell out the AGI abbreviation (either in the title or in a footnote). Finally prefer using “laboratory pathogen testing” in the place of the vaguer “laboratory perming test for pathogens”

- L174-178: please revise to “The estimated cases in the surveillance area during the 12-month study 1,349 cases (95% CI…) of salmonellosis and 3,408 cases (95% CI…) of *V. parahaemolyticus* infection (Table 3). Considering the population of the surveillance area, the annual incidence estimated by the model developed herein was 101 (95% CI…) cases for salmonellosis abd 256 (95% CI…) cases for *V. parahaemolyticus* infection per 100,000 population”

- Table 3: Revise the title to “Estimated health burden of non-typhoidal salmonellosis and Vibrio parahaemolyticus infection in Shanghai, east Chine, 2010-2011. Also make the following changes:

1.  Non-typhoidal *Salmonella enterica *(in 1^st^ column)

2. Estimated foodborne illness cases per 100,000 population (95% CI) (in 4^th^ and 6^th^ column)

3. Reported foodborne illness outbreak cases per 100,000 population (in 7^th^ column)

General comment: avoid using the general term “illness” and try to be consistent and specific by using the term “foodborne illness”; also choose whether you prefer to use “illness” or “disease” and be consistent throughout the manuscript.

- L184: correct to “from a population survey”

- L184-185: revise to “…surveillance, the burden of foodborne gastroenteritis in Shanghai was estimated to be 13,310…”

- L187: revise to “This indicates that AGI caused by these two pathogens poses a substantial burden…”

- L197: revise to “…as determined from a literature review…”

- L210: what about the distinction between outbreak and sporadic cases? Is this possible based on the analysis performed in the present study? I think that a pertinent comment would add value to the Discussion section of the manuscript.

- L215: revise to “…the China National Center…”

- L218” revise to “…estimating the burden…”

- L223-226: please consider revising to “In order to more accurately determine and prioritize food safety issue in China, and to evaluate and quantify the burden of foodborne diseases outbreaks, it is necessary…

- L241-245: please revise to “In conclusion, the estimated large number of salmonellosis and *V. parahaemolyticus* infection cases occurring every year in the surveillance area, indicates that these two pathogens pose a substantial health burden in Shanghai, east China, After considering the differences among distinct pathogens, these methods can also be applied in a similar manner to assess the burden of additional foodborne pathogens”

- L246: change “infection” to “infections”

- L248: revise to “…in order to provide improved data support…”

- References: please cross-check references with regard to accuracy and conformance to the PLOS ONE format/style

- Figure 2: correct the legends’ text to “Non-typhoidal *Salmonella enterica*” and “*Vibrio parahaemolyticus*” 

We look forward to receiving your revised manuscript.

Kind regards,

Alexandra Lianou, Ph.D.

Academic Editor

PLOS ONE

Journal Requirements:

3. In your Methods section, please provide additional information about the demographic details of your participants. Please ensure you have provided sufficient details to replicate the analyses such as: a)  a description of any inclusion/exclusion criteria that were applied to participant inclusion in the analysis and b) a table of relevant demographic details.

4. You indicated that you had ethical approval for your study. In your Methods section, please ensure you have also stated whether you obtained consent from parents or guardians of the minors included in the study or whether the research ethics committee or IRB specifically waived the need for their consent.

5. In the ethics statement in the manuscript and in the online submission form, please provide additional information about the patient records used in the retrospective hospital surveillance arm of your study. Specifically, please ensure that you have discussed whether all data were fully anonymized before you accessed them and/or whether the IRB or ethics committee waived the requirement for informed consent. If patients provided informed written consent to have data from their medical records used in research, please include this information.

Reviewers' comments:

Reviewer's Responses to Questions

**Comments to the Author**

1. Is the manuscript technically sound, and do the data support the conclusions?

Reviewer #1: Yes

2. Has the statistical analysis been performed appropriately and rigorously? 

Reviewer #1: Yes

3. Have the authors made all data underlying the findings in their manuscript fully available?

Reviewer #1: Yes

4. Is the manuscript presented in an intelligible fashion and written in standard English?

Reviewer #1: Yes

5. Review Comments to the Author

Reviewer #1: 1. The Introduction provides justification for conducting a national health burden assessment but does not explain why a local (Shanghai) health burden assessment was conducted. Please explain why a local and not a national health burden assessment was conducted and why Salmonella and Vibrio parahaemolyticus were selected for the assessment.

2. What is the difference between a primary, secondary, and tertiary hospital? Other readers may have similar questions. Therefore, it might be a good idea to include the answer to this question in the manuscript.

3. Is the testing sensitivity the same as the false negative rate? It seems that a range from 50 to 100% is not realistic because no test is perfect and a test with such a low sensitivity of 50% would not be used. It would be a good idea to better explain the basis for these estimates of uncertainty.

4. When I multiply the most likely values for M1, M2, M3, and M4 for the pert distributions in Table 1, I get 51 but the text says the overall multiplier is 71. Why are these values not similar? Other readers may have the same question. Perhaps it would be good to explain in more detail how 71 and its 95% CI were obtained.

5. It is my understanding, based on human feeding trials (McCullough & Eisele, 1951a, 1951b, 1951c, 1951d), that an infection occurs when a patient is shedding the pathogen but not showing symptoms of disease, whereas an illness occurs when a patient is shedding the pathogen and showing symptoms of the disease. In the present study, the incidence of people showing symptoms of gastrointestinal disease was a basis for the calculation of health burden. Thus, the health burden assessment was for illness and not infection. Yet, throughout the paper both terms are used interchangeably, which is a bit confusing. To do a health burden assessment for infection, data would be needed for the incidence of people that test positive for the pathogen but do not show symptoms of illness. That kind of data was not collected in the present study. Thus, I think that it is not appropriate to talk about a health burden assessment for infection when it is actually a health burden assessment for illness.

6. The sentence starting on line 153 seems to be missing its beginning. Thus, its meaning is not clear. Please clarify this sentence.

7. When a person is exposed to a foodborne pathogen, their peak response falls on a continuum from no infection to infection (asymptomatic) to illness (symptomatic) to severe illness (hospital) to death. Where the peak response falls on this continuum depends on the outcome of the interaction between the pathogen, host, and food (disease triangle). Consequently, if two communities had the same illness rate but one community had more high-risk individuals, the illness rate would be a poor indicator of health burden because the severity of illness would higher in the community with more high-risk individuals. Therefore, I think the current manuscript could be improved by estimating a health burden that takes severity of illness into account.

8. Overall, I think this is a very good paper that will be a good addition to the scientific literature. My comments are mainly suggestions that if adopted could improve the manuscript. My main suggestions are to justify conducting a local health burden assessment, to focus the health burden assessment on illness and not infection, and to consider severity of illness in the prediction of health burden.

I hope this review is helpful,

References

McCullough, N. B., & Eisele, C. W. (1951a). Experimental human salmonellosis. I. Pathogenicity of strains of Salmonella meleagridis and Salmonella anatum obtained from spray-dried whole egg. Journal of Infectious Disease, 88, 278-289.

McCullough, N. B., & Eisele, C. W. (1951b). Experimental human salmonellosis. III. Pathogenicity of strains of Salmonella newport, Salmonella derby, and Salmonella bareilly obtained from spray-dried whole egg. Journal of Infectious Disease, 89, 209-213.

McCullough, N. B., & Eisele, C. W. (1951c). Experimental human salmonellosis. IV. Pathogenicity of strains of Salmonella pullorum obtained from spray-dried whole egg. Journal of Infectious Disease, 89, 259-265.

McCullough, N. B., & Eisele, C. W. (1951d). Experimental human salmonellosis: II. Immunity studies following experimental illness with Salmonella meleagridis and Salmonella anatum. Journal of Immunology, 66, 595-608.

6. PLOS authors have the option to publish the peer review history of their article (what does this mean?). If published, this will include your full peer review and any attached files.

Reviewer #1: No

---

## [Author Response · Author response to Decision Letter 0]

2 Oct 2020

PONE-D-20-22153

The Human Health Burden of non-typhoidal Salmonella enterica and Vibrio parahaemolyticus Foodborne Gastroenteritis in Shanghai, East China

PLOS ONE

Editor’s comments

Although the Academic Editor is not an epidemiologist, raised comments regarding the overall approach of the presented study from a food microbiology perspective, its scientific soundness as well as the quality of the manuscript are summarized below. According to the editor’s opinion, the manuscript should be subject to minor revisions before its publication in PLOS ONE is considered. Beyond the more specific comments that are following, a general a comment is an overall cross-check with regard to English grammar/syntax (maybe a review by an English expert/native speaker should be considered). 

Response: I have revised the manuscript via a native English-speaking expert.

- Title: both in the manuscript’s title and throughout the manuscript, please use the species name “Salmonella enterica” (or S. enterica after first mentioning) instead of the genus name (i.e. Salmonella). The genus Salmonella includes two species (i.e. Salmonella enterica and Salmonella bongori) and only S. enterica is regarded as a foodborne pathogen for humans.

Response: Accepted and revised.

- L24-25: please revise to “…cases for non-typhoidal salmonellosis and V. parahaemolyticus infection, respectively, illustrating that bacterial gastroenteritis due to these two pathogens poses a substantial health burden”

Response: Accepted and revised as following:

Annual incidence per 100,000 population in Shanghai was estimated as 48 (95% CI 24–83) and 183 (95% CI 93–317) cases for foodborne non-typhoidal salmonellosis and V. parahaemolyticus gastroenteritis, respectively, illustrating that bacterial gastroenteritis due to these two pathogens poses a substantial health burden. (L23-27)

- L27: revise to “…and the data actually reported for foodborne diseases…”

Response: Accepted and revised.

- L45: I would suggest using “relative” instead of “proportionate”

Response: Accepted and revised. (L47)

- L48: please correct to “the Netherlands”

Response: Accepted and revised. (L50)

- L53: please revise to “…for precisely estimating foodborne disease burden and monitoring the impact…”

Response: Accepted and revised. (L56)

- L58-59: please revise to “…among patients provides important basic data for estimating foodborne disease burden”

Response: Accepted and revised. (L62)

- L64: the phrase “foodborne pathogenic infections” is rather specific for such a general statement; I would suggest using “foodborne illness severity” instead.

Response: Accepted and revised. (L67)

- L92: correct to “eight sentinel hospitals”

Response: Accepted and revised. (L103)

- L94-95: please correct to “Rectal swabs or stool specimens from patients with diarrhea were collected from the sentinel hospitals…”

Response: Accepted and revised. (L109)

- L96: revise to “non-typhoidal S. enterica and V. parahaemolyticus by the laboratories of the Chinese Center for Disease Control and Prevention”; in general, be consistent using the same single (and not multiple) terms throughout the manuscript, e.g. “non-typhoidal” (instead of “non-typhoid” or “nontyphoidal”), “foodborne” (instead of “food-borne”), etc.

Response: Accepted and revised. (L101-111)

- L98: please correct to “…were representative of the community samples”

Response: Accepted and revised. (L113)

- L99-103: revise to “For S. enterica detection, specimens were enriched in selenite broth, followed by surface plating (or plating) on Bismuth sulfite agar and xylose-lysine-desoxycholate (XLD) agar. With reference to V. parahaemolyticus detection, specimens were enriched in alkaline peptone water, followed by surface plating (or streaking) on thiosulfate citrate bile salts sucrose agar or CHROMagar Vibrio agar”. Moreover, were specific ISO protocols applied for these selective enrichment procedures? If so, please refer to the specific protocols (as in-text citations with the corresponding additions in the references’ list). 

Response: Accepted and revised as following:

For S. enterica detection, specimens were enriched in selenite broth, followed by surface plating (or plating) on Bismuth sulfite agar and xylose-lysine-desoxycholate (XLD) agar (or Hektoen enteric agar, CHROMagar Salmonella agar) [23]. With reference to V. parahaemolyticus detection, specimens were enriched in alkaline peptone water, followed by surface plating (or streaking) on thiosulfate citrate bile salts sucrose agar or CHROMagar Vibrio agar [24].

- L107-113: it would be better for the reader (particularly a non-expert in epidemiology) if definitions of M1, M2 and M3 are provided in brief

Response: Accepted and revised. (L125-139)

- L111: please change to “Then, the number of submitted stool specimens was divided by…”

Response: Accepted and revised. (L128)

- L112: by “laboratory performing test for pathogens” you refer to “laboratory pathogen testing”? If so, consider using the second (simpler) term both here and wherever else applicable in the manuscript’s text.

Response: Accepted and revised. (L125, 126, 197)

- L114: change to “It was assumed that the laboratory testing sensitivity was 87.5%...”; in general, use third person (and not “we”) when providing descriptions pertinent to the Materials and Methods and/or Results section of the manuscript.

Response: Accepted and revised. (L127-128)

- L117: please revise to “Figure 1 shows the pyramid of the burden of pathogen-specific foodborne gastroenteritis”

Response: Accepted and revised. (L141-142)

- L118-119: please revise to “…were modeled using the Pert distribution. The estimation model was developed using @RISK…”

Response: Accepted and revised. (L143-144)

- L120: please revise to “The number of positive specimens tested was multiplied with M to estimate the number of sportive samples in the surveillance area”

Response: Accepted and revised. (L145-146)

- L123: please correct to “…were assessed”

Response: Accepted and revised. (L149)

- L124: revise to “Pyramid of the burden of pathogen-specific gastroenteritis”. Also, is this the appropriate place in the manuscript’s text for a figure caption (and the same applies for Fig. 2 caption in L157-158)?

Response: Accepted and revised. (L150)

- L125-126: please revise to “…non-typhoidal salmonellosis and Vibrio parahaemolyticus infection…”; please use the same phrase when referring to the infection (gastroenteritis) caused by these two pathogens throughout the manuscript for simplification and consistency reasons.

Response: Accepted and revised. (L151-152)

- Table 1: in the last line (referring to M4) and the last column (referring to the distribution) please use a decimal digit for all three numbers (namely, 1.0, 1.1 and 2.0).

Response: Accepted and revised. 

- L129-135: please revise as following recommended: “Since there are no data on the proportion…on data published by Hald et al. [23]…and were multiplied by the number of illnesses to obtain foodborne non-typhoidal salmonellosis [23], while the proportion based on US research was used to obtain…The estimated pathogen-specific numbers of foodborne illness cases were compared with the routinely reported numbers of foodborne outbreak cases so as to calibrate the surveillance data”. Please keep in mind that “data” is plural (singular form the Latin word “datum”), and proceed with all required grammar corrections when using this term throughout the manuscript.

Response: Accepted and revised as following. 

Since there are no data on the proportion of foodborne pathogen-specific transmission in China, the proportions were based on data published by Hald et al. for WPR B region (including China) and were multiplied by the number of diseases to obtain foodborne non-typhoidal salmonellosis [26], while the proportion based on US research was used to obtain foodborne V. parahaemolyticus gastroenteritis [2]. The estimated pathogen-specific numbers of foodborne disease cases were compared with the routinely reported numbers of foodborne outbreak cases so as to calibrate the surveillance data [8]. (L156-163)

- L138: change to “…the National Institute for Nutrition and Food Safety…”

Response: Accepted and revised. (L164)

- L149-150: please revise to “…36.9% (95% CI 26.5-47.2) visited a doctor, and among them 34.4% (95% CI 17.0-51.9) submitted a stool specimen”

Response: Accepted and revised. (L179-180)

- L155-156: please correct to “…(Fig. 2), while more information regarding the two studied bacterial species was not collected”

Response: Accepted and revised. (L191-192)

- L157-158: please revise the Figure 2 caption to “Month-wise isolation rate for non-typhoidal Salmonella enterica and Vibrio parahaemolyticus in Shanghai, east China, 2010-2011”

Response: Accepted and revised. (L194-195)

- L165: please correct to “Among people seeking medical care, 34.4%...”

Response: Accepted and revised. (L202)

- L168: please correct to “Considering that the number of stool specimens tested in the surveillance area was 4,548…”

Response: Accepted and revised. (L205)

- L170: change to “…laboratory confirmed salmonellosis or V. parahaemolyticus infection…”; also to what does “respectively” refer to? (a single percentage, namely 71%, is mentioned herein).

Response: Accepted and revised. (L208)

- Table 2: I would suggest moving the percentage symbol (%) in the first (from the second) column of the table in parentheses (wherever applicable), e.g., “AGI incidence per person-year (%)”. Moreover, spell out the AGI abbreviation (either in the title or in a footnote). Finally prefer using “laboratory pathogen testing” in the place of the vaguer “laboratory perming test for pathogens”

Response: Accepted and revised. (L211)

- L174-178: please revise to “The estimated cases in the surveillance area during the 12-month study 1,349 cases (95% CI…) of salmonellosis and 3,408 cases (95% CI…) of V. parahaemolyticus infection (Table 3). Considering the population of the surveillance area, the annual incidence estimated by the model developed herein was 101 (95% CI…) cases for salmonellosis abd 256 (95% CI…) cases for V. parahaemolyticus infection per 100,000 population”

Response: Accepted and revised. (L199-205)

The estimated cases in the surveillance area during the 12-month study were 1,121 cases (95% CI570–1,938) of salmonellosis and 2,832 cases (95% CI 1,440–4,896) of V. parahaemolyticus gastroenteritis (Table 4). Considering the population of the surveillance area, the annual incidence estimated by the model developed herein was 84 (95% CI 43–146) cases for salmonellosis and 213 (95% CI 108–368) cases for V. parahaemolyticus gastroenteritis per 100,000 population. Annual incidence of foodborne gastroenteritis per 100,000 population in Shanghai was estimated as 48 (95% CI (24–83) for non-typhoidal salmonellosis and 183 (95% CI93–317) cases for V. parahaemolyticus gastroenteritis.

- Table 3: Revise the title to “Estimated health burden of non-typhoidal salmonellosis and Vibrio parahaemolyticus infection in Shanghai, east Chine, 2010-2011. Also make the following changes:

1. Non-typhoidal Salmonella enterica (in 1st column)

2. Estimated foodborne illness cases per 100,000 population (95% CI) (in 4th and 6th column)

3. Reported foodborne illness outbreak cases per 100,000 population (in 7th column)

Response: Accepted and revised. (L207-208)

General comment: avoid using the general term “illness” and try to be consistent and specific by using the term “foodborne illness”; also choose whether you prefer to use “illness” or “disease” and be consistent throughout the manuscript.

Response: Accepted and revised. 

- L184: correct to “from a population survey”

Response: Accepted and revised. (L228)

- L184-185: revise to “…surveillance, the burden of foodborne gastroenteritis in Shanghai was estimated to be 13,310…”

Response: Accepted and revised. (L228-229)

- L187: revise to “This indicates that AGI caused by these two pathogens poses a substantial burden…”

Response: Accepted and revised. (L232)

- L197: revise to “…as determined from a literature review…”

Response: Accepted and revised. (L242)

- L210: what about the distinction between outbreak and sporadic cases? Is this possible based on the analysis performed in the present study? I think that a pertinent comment would add value to the Discussion section of the manuscript.

Response: Accepted and added sentences as following. (L252-258)

Diseases are divided into outbreak and sporadic forms. When designing and implementing pathogen-specific foodborne disease burden studies, it is necessary to consider the distinction between outbreak and sporadic cases. As shown by data from Japan and the United States, Salmonella and V. parahaemolyticus infections are mainly sporadic, and outbreaks are less common [2, 15]. Because outbreaks and sporadic diseases have similar case characteristics [31], the analysis of the estimated disease burden based on laboratory-confirmed sporadic cases adopted in this study is feasible.

- L215: revise to “…the China National Center…”

Response: Accepted and revised. (L269)

- L218” revise to “…estimating the burden…”

Response: Accepted and revised. (L271)

- L223-226: please consider revising to “In order to more accurately determine and prioritize food safety issue in China, and to evaluate and quantify the burden of foodborne diseases outbreaks, it is necessary…

Response: Accepted and revised. (L277-278)

- L241-245: please revise to “In conclusion, the estimated large number of salmonellosis and V. parahaemolyticus infection cases occurring every year in the surveillance area indicates that these two pathogens pose a substantial health burden in Shanghai, east China, After considering the differences among distinct pathogens, these methods can also be applied in a similar manner to assess the burden of additional foodborne pathogens”

Response: Accepted and revised. (L328-333)

In conclusion, the estimated large number of salmonellosis and V. parahaemolyticus gastroenteritis cases occurring every year in the surveillance area, indicates that these two pathogens pose a substantial health burden in Shanghai, east China. After considering the differences among distinct pathogens, these methods can also be applied in a similar manner to assess the burden of additional foodborne pathogens.

- L246: change “infection” to “infections”

Response: Accepted and revised. (L334)

- L248: revise to “…in order to provide improved data support…”

Response: Accepted and revised. (L336)

- References: please cross-check references with regard to accuracy and conformance to the PLOS ONE format/style

Response: Accepted and revised. (L262, L271)

- Figure 2: correct the legends’ text to “Non-typhoidal Salmonella enterica” and “Vibrio parahaemolyticus” 

Response: Accepted and revised. (L194)

Response: Accepted and include a questionnaire as Supporting Information.

3. In your Methods section, please provide additional information about the demographic details of your participants. Please ensure you have provided sufficient details to replicate the analyses such as: a) a description of any inclusion/exclusion criteria that were applied to participant inclusion in the analysis and b) a table of relevant demographic details.

Response: Accepted and added a sentence as following:

but excluding those persons who reported their symptoms of diarrhea or vomiting to be due to non-infectious causes such as Crohn’s disease, irritable bowel syndrome, colitis, diverticulitis of large intestine, pregnancy, excess alcohol, chemotherapy/radiotherapy, drugs, or food allergy. (L92-95)

Also, a table of relevant demographic details was added, see L183.

4. You indicated that you had ethical approval for your study. In your Methods section, please ensure you have also stated whether you obtained consent from parents or guardians of the minors included in the study or whether the research ethics committee or IRB specifically waived the need for their consent.

Response: Accepted and added sentences as following:

Written and informed consent was received from all respondents and parents or guardians of the minors prior to the interview. (L86-87)

The ethics committee waived the requirement for informed consent from patients with diarrhea. (L167-168)

5. In the ethics statement in the manuscript and in the online submission form, please provide additional information about the patient records used in the retrospective hospital surveillance arm of your study. Specifically, please ensure that you have discussed whether all data were fully anonymized before you accessed them and/or whether the IRB or ethics committee waived the requirement for informed consent. If patients provided informed written consent to have data from their medical records used in research, please include this information.

Response: Accepted and added sentences as following:

No detailed personal information was collected on patients with diarrhea. (L192-193)

Reviewer #1: 

1. The Introduction provides justification for conducting a national health burden assessment but does not explain why a local (Shanghai) health burden assessment was conducted. Please explain why a local and not a national health burden assessment was conducted and why Salmonella and Vibrio parahaemolyticus were selected for the assessment.

Response: Accepted and added sentences as following:

From 2010 to 2011, the Shanghai Municipal Center for Disease Control and Prevention launched a pilot project for active surveillance of foodborne diseases. Non-typhoidal Salmonella enterica and Vibrio parahaemolyticus are the most common bacteria that cause foodborne disease outbreaks in China [8]. (L67-70)

2. What is the difference between a primary, secondary, and tertiary hospital? Other readers may have similar questions. Therefore, it might be a good idea to include the answer to this question in the manuscript.

Response: Accepted and added sentences as following:

A primary hospital was defined as a community hospital that provided primary health services; a secondary hospital was defined as a local hospital that provided comprehensive health services; and a tertiary hospital was defined as a regional hospital that provided comprehensive and specialized health services [22]. (L105-109)

3. Is the testing sensitivity the same as the false negative rate? It seems that a range from 50 to 100% is not realistic because no test is perfect and a test with such a low sensitivity of 50% would not be used. It would be a good idea to better explain the basis for these estimates of uncertainty.

Response: Accepted and revised as following:

According to Proficiency testing program in Guangdong province, the Salmonella isolation sensitivity rate of the laboratories was 87.5% (L134-135)

4. When I multiply the most likely values for M1, M2, M3, and M4 for the pert distributions in Table 1, I get 51 but the text says the overall multiplier is 71. Why are these values not similar? Other readers may have the same question. Perhaps it would be good to explain in more detail how 71 and its 95% CI were obtained.

Response: Accepted and revised as following:

The above mentioned multipliers were multiplied to estimate the multiplier for these surveillance artifacts (Multiplier Total, MT). (L140-141)

For each person with laboratory confirmed salmonellosis or V. parahaemolyticus gastroenteritis there were 59 (95% CI30–102) infected persons in the community (MT). (L194-195)

5. It is my understanding, based on human feeding trials (McCullough & Eisele, 1951a, 1951b, 1951c, 1951d), that an infection occurs when a patient is shedding the pathogen but not showing symptoms of disease, whereas an illness occurs when a patient is shedding the pathogen and showing symptoms of the disease. In the present study, the incidence of people showing symptoms of gastrointestinal disease was a basis for the calculation of health burden. Thus, the health burden assessment was for illness and not infection. Yet, throughout the paper both terms are used interchangeably, which is a bit confusing. To do a health burden assessment for infection, data would be needed for the incidence of people that test positive for the pathogen but do not show symptoms of illness. That kind of data was not collected in the present study. Thus, I think that it is not appropriate to talk about a health burden assessment for infection when it is actually a health burden assessment for illness.

Response: Accepted and the health burden assessment was for disease.

6. The sentence starting on line 153 seems to be missing its beginning. Thus, its meaning is not clear. Please clarify this sentence.

Response: Accepted and revised as following:

Between July 2010 and June 2011, a total of 4,568 patients with diarrhea presented to a hospital participating in surveillance, among which, 4,548 swab/stool specimen were collected and tested. (L174-176)

7. When a person is exposed to a foodborne pathogen, their peak response falls on a continuum from no infection to infection (asymptomatic) to illness (symptomatic) to severe illness (hospital) to death. Where the peak response falls on this continuum depends on the outcome of the interaction between the pathogen, host, and food (disease triangle). Consequently, if two communities had the same illness rate but one community had more high-risk individuals, the illness rate would be a poor indicator of health burden because the severity of illness would higher in the community with more high-risk individuals. Therefore, I think the current manuscript could be improved by estimating a health burden that takes severity of illness into account.

Response: Accepted and revised as following:

When a person is exposed to a foodborne pathogen, their peak response falls on a continuum from no infection to infection (asymptomatic) to disease (symptomatic) to severe disease (hospital) to death. Where the peak response falls on this continuum depends on the outcome of the interaction between the pathogen, host, and food (disease triangle). Consequently, if two communities had the same disease rate but one community had more high-risk individuals, the disease rate would be a poor indicator of health burden because the severity of disease would higher in the community with more high-risk individuals. Therefore, it is necessary to collect data on severity of disease in future researches, and the current assessment could be improved by estimating the health burden that takes severity of disease into account. (L317-326)

---

## [Editor Report · Decision Letter 1]

28 Oct 2020

The Human Health Burden of non-typhoidal Salmonella enterica and Vibrio parahaemolyticus Foodborne Gastroenteritis in Shanghai, East China

PONE-D-20-22153R1

Dear Dr. Chen,

We’re pleased to inform you that your manuscript has been judged scientifically suitable for publication and will be formally accepted for publication once it meets all outstanding technical requirements.

Kind regards,

Alexandra Lianou, Ph.D.

Academic Editor

PLOS ONE

Additional Editor Comments (optional):

All comments raised have been sufficiently addressed in the revised manuscript.
---

## [Editor Report · Acceptance letter]

4 Nov 2020

PONE-D-20-22153R1 

The Human Health Burden of non-typhoidal *Salmonella enterica* and *Vibrio parahaemolyticus* Foodborne Gastroenteritis in Shanghai, East China 

Dear Dr. Chen:

I'm pleased to inform you that your manuscript has been deemed suitable for publication in PLOS ONE. Congratulations! Your manuscript is now with our production department. 

Kind regards, 

on behalf of

Dr. Alexandra Lianou 

Academic Editor

PLOS ONE